# Sulfate geoengineering: a review of the factors controlling the needed injection of sulfur dioxide

Daniele Visioni[1,2], Giovanni Pitari[1], and Valentina Aquila[3,4]

[1]Department of Physical and Chemical Sciences, Universitá dell'Aquila, 67100 L'Aquila, Italy
[2]CETEMPS, Universitá dell'Aquila, 67100 L'Aquila, Italy
[3]GESTAR/Johns Hopkins University, Department of Earth and Planetary Science, 3400 N Charles Street, Baltimore, MD 21218, USA
[4]NASA Goddard Space Flight Center, Code 614, Greenbelt, MD 20771, USA

*Correspondence to:* Daniele Visioni (daniele.visioni@aquila.infn.it)

**Abstract.**

Sulfate geoengineering has been proposed as an affordable and climate-effective mean to temporarily offset the warming produced by the increase of well mixed greenhouse gases (WMGHG). This technique would likely have to be applied during and after global intergovernmental measures on emissions of WMGHGs are implemented, in order to achieve surface temperature stabilization. The direct radiative effects of sulfur injection in the tropical lower stratosphere can be summarized as increasing shortwave scattering with consequent tropospheric cooling and increasing longwave absorption with stratospheric warming. Indirect radiative effects are related to induced changes in the ozone distribution, stratospheric water vapor abundance, formation and size of upper tropospheric cirrus ice particles and lifetime of longlived species, namely $CH_4$ in connection with OH changes through several photochemical mechanisms. Direct and indirect effects of sulfate geoengineering both concur to determine the atmospheric response. A review of previous studies on these effects is presented here, with an outline of the important factors that control the amount of sulfur dioxide to be injected in an eventual realization of the experiment. However, we need to take into account that atmospheric models used for these studies have shown a wide range of climate sensitivity and differences in the response to stratospheric volcanic aerosols. In addition, large uncertainties exist in the estimate of some of these aerosol effects.

## 1 Introduction

The overwhelming evidence of a surface warming caused by the anthropogenic increase in greenhouse gases (GHG) has forced the scientific community to look for methods of mitigating and possibly reversing this trend (IPCC (2007)). Such a need is made even more pressing if we look at the projections for the next century. The Intergovernmental Panel on Climate Change (IPCC) has built various Representative Concentration Pathways (RCPs) predicting future anthropogenic emissions (greenhouse gases, anthropogenic aerosols, short lived gas species etc.) and assessed the effect of such scenarios on the Earth's climate using a series of multi-model experiments (CMIP5) (Taylor et al. (2012)). The main result is the agreement among most models on a warming of the Earth's surface ranging from a 1 K increase by 2100 for the most optimistic scenario (RCP2.6, with near-

constant emissions between 2020 and 2100) to a 3.7 K increase for the least optimistic scenario (RCP8.5, with most developing countries increasing their emissions sensibly) (Meinshausen et al. (2011)). These forecasts tell us that, even with the most optimistic emission scenario, a sudden reversing of the temperature trend is not expected (IPCC (2007); Nordhaus (2007)).

In order to mitigate the effects that such a warming would have on the climate of our planet, some methods have been proposed
to balance out the direct effects of GHG, generally known under the name of climate engineering or geoengineering. Geoengineering methods have to be carefully evaluated on four grounds: effectiveness (the potential for the proposed method to work), affordability, timeliness (how long it would take to deploy it and how fast would it work) and safety (the risks linked with the deployment of the method). Such geoengineering methods would need to be applied during and after global intergovernmental measures on GHG emissions are implemented, in order to achieve surface temperature stabilization (Sanderson et al. (2016);
Tilmes et al. (2016)). These methods can be divided into two large groups: the first group is composed of carbon dioxide removal techniques, whose aim is to directly reduce the amount of carbon dioxide in the atmosphere by means such as afforestation, atmospheric $CO_2$ scrubbers, in-situ carbonation of silicate over land, and fertilization and alkalinity enhancements over the oceans. The second group, in which the method we will be studying further on is situated, is the one known under the term Solar Radiation Management (SRM) techniques, whose aim is to decrease the amount of incoming radiation on the Earth
surface: among those we find surface albedo increase, cloud albedo enhancement, space-based reflectors, and stratospheric aerosol injection, also called sulfate engineering (CEC (2014)).

Sulfate geoengineering (SG) prescribes the sustained injection of sulfur dioxide ($SO_2$) in the tropical lower stratosphere, originally proposed by Budyko (2013) and further developed by Crutzen (2006). Under the international modeling project GeoMIP (Geoengineering Model Intercomparison Project; Robock et al. (2011); Kravitz et al. (2011); Kravitz et al. (2012); Kravitz
et al. (2013)) chemistry-climate models and atmosphere-ocean coupled models have been used to explore the radiative, chemical and dynamical modification of climate by $SO_2$ injection. Several studies were conducted to compare a control simulation ensemble under the IPCC scenario RCP4.5 (Taylor et al. (2012)) and a sulfate geoengineering simulation. In this review we summarize the direct and indirect climate effects of a constant stratospheric injection of $SO_2$, such as the one prescribed by the GeoMIP experiment G4 (Pitari et al. (2014); Aquila et al. (2014a)) and in earlier studies (Rasch et al. (2008); Tilmes et al.
(2009); Tilmes et al. (2012)) or of a time-varying $SO_2$ injection, such as in the GeoMIP experiment G3 (Pitari et al. (2014)). In this case the amount of the injected $SO_2$ changes year-by-year in order to keep the top-of-atmosphere (TOA) radiative balance constant (Robock et al. (2011); Kravitz et al. (2011)). The G4-type approach (even if with different amounts of constant $SO_2$ injection) has been used and documented in a wider number of studies (see also Heckendorn et al. (2009); Niemeier et al. (2011); English et al. (2012); Niemeier et al. (2013); Niemeier and Timmreck (2015); Tilmes et al. (2015)).
The direct effect of an injection of $SO_2$ is an increase in the local concentration of optically active $H_2O$-$H_2SO_4$ aerosol particles in the lower stratosphere. These particles increase the amount of back-scattered solar radiation, resulting in less radiation arriving at the Earth's surface, thus cooling the whole troposphere. The idea itself of sulfate geoengineering comes from the observation of various explosive volcanic eruptions over the last century, which injected large amounts of sulfur in the lower stratosphere over a very short amount of time and whose direct impact on the global mean surface temperature has been known

for some time (Robock and Mao (1995)).

## 2 Review of radiative forcing effects

### 2.1 Direct forcing of stratospheric sulfate

The underlying physical processes behind the injection of $SO_2$ into the atmosphere have been widely studied thanks to the various explosive volcanic eruptions of the $20^{th}$ century. For instance, after the Mount Pinatubo eruption of June 1991, when 7 to 10 Tg-S were injected into the stratosphere (Read et al. (1993); Krueger et al. (1995)), a sharp reduction in the TOA net radiative flux was observed in the year following the eruption ($\sim$2.5 W/m$^2$) (Stowe et al. (1992)), as well as a significant drop in global surface temperatures of about 0.5 K (Dutton and Christy (1992)). This was calculated as a monthly mean for

September 1992, compared to pre-Pinatubo levels. However, more recent results with detrended analyses (Canty et al. (2013)) have shown that the Pinatubo volcanic impact on surface temperatures was probably overestimated by about a factor of 2, with a cooling estimate of 0.14 K and 0.32 K, globally and over land, respectively.

These effects can be explained by $SO_2$ oxidation into $SO_4$ followed by the formation of $H_2O$-$H_2SO_4$ supercooled liquid droplets, which create an optically thick cloud that reflects part of the incoming solar radiation. This results in a surface cool-

ing and a local stratospheric warming. The stratospheric warming is due to changes in diabatic heating rates produced by aerosol absorption of solar near infrared and planetary radiation and by the ozone absorption of the additional UV radiation scattered by the volcanic aerosols (Pitari (1993)).

When considering the effects of the proposed injection of sulfur into the atmosphere, however, a series of factors must be taken into account, complicating the analogy between this kind of geoengineering experiments and volcanic eruptions. Obviously,

the amount of sulfur and the height and latitude at which it is injected in a geoengineering experiment all play a prominent role in its related effects. Some recent papers, such as English et al. (2012) and Niemeier and Timmreck (2015) analyzed a series of geoengineering experiments accounting for the different factors previously mentioned. Their results show that the relation between injected $SO_2$ and the resulting sulfate mass burden is non-linear, with larger injection rates producing a lower efficiency of SG. This is due to the fact that injections of larger amounts of $SO_2$ lead to the formation of larger aerosol particles

by gas condensation, which are rapidly removed from the stratosphere by gravitational settling (see Fig. 1, with calculated vertical profiles of the aerosol effective radius).

Aside from the reduction in the aerosol lifetime, the size of the produced aerosol particles also influences the amount of scattered radiation, because the sulfate scattering efficiency peaks at a particle radius of around 140 nm and decreases as aerosols become larger (English et al. (2012)). The highest burden to injection ratio is achieved for stratospheric injections between

30N and 30S (English et al. (2012)), because gas condensation and particle coagulation are both reduced with $SO_2$ injection spanning over a broader latitude. The altitude also plays a significant role in determining the aerosol lifetime, due to a faster sedimentation removal in the upper troposphere (UT) when the sulfur injection is localized closer to the tropical tropopause

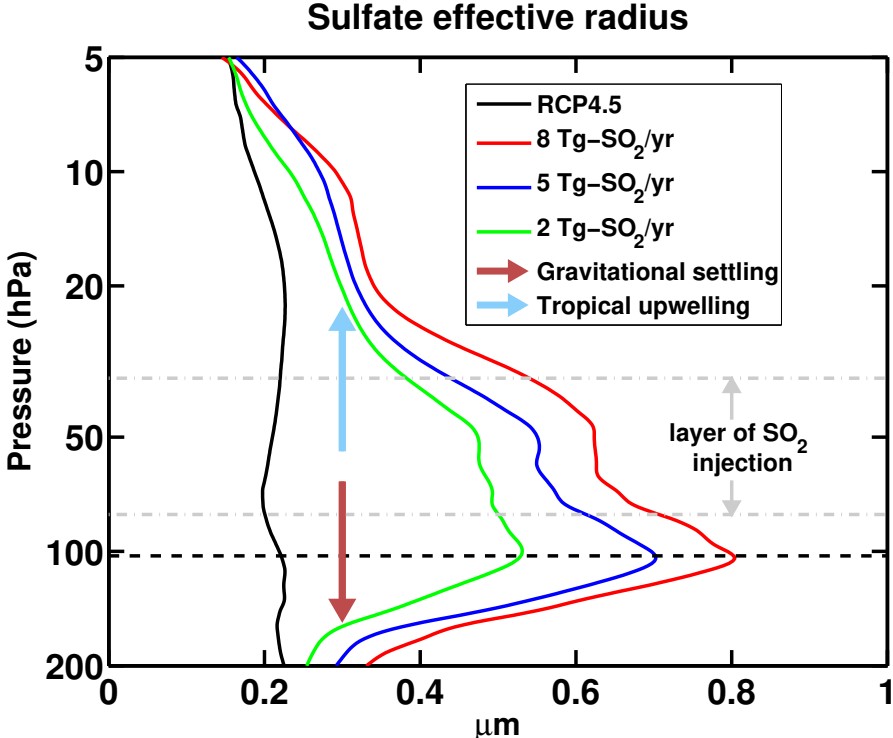

**Figure 1.** Annual averaged vertical profiles of aerosol effective radius ($\mu$m) in the tropical stratosphere (25S-25N), with increasing geoengineering injection of $SO_2$ (see legend). The heavy dashed line indicates the mean tropical tropopause. Profiles are calculated in the University of L'Aquila Chemistry-Climate Model (ULAQ-CCM), which includes explicit gas-particle conversion and aerosol microphysics (Pitari et al. (2014)).

layer (TTL) (Aquila et al. (2014a)).

As shown in Pitari et al. (2014), the injection of 5 Tg-$SO_2$/yr produces, according to the models used in the experiment G4, a net TOA radiative forcing (RF) of -1.54 W/m$^2$, -1.27 W/m$^2$, -1.31 W/m$^2$ and -0.73 W/m$^2$, for ULAQ-CCM, GEOSCCM, GISS-E2-R and MIROC-ESM-CHEM, respectively (Pitari et al. (2014) for model description and details). The different results are mainly dependent on the (calculated, or imposed in one case) different aerosol optical depth (AOD) and size distribution among models. It should also be considered that, in general, even with the same AOD distribution, models may produce different radiative responses depending on the adopted radiation scheme (Neely et al. (2016)). Other RF values are available from literature, for a variety of conditions of sulfur injection (amount and altitude, mainly). With a linear scaling to 5 Tg-$SO_2$/yr (in case of different injection values), we get the following values: -1.13 W/m$^2$ (Heckendorn et al. (2009)); -1.17 W/m$^2$ (Niemeier et al. (2011)); -1.53 W/m$^2$ (Kuebbeler et al. (2012)); -1.4 W/m$^2$ and -1.0 W/m$^2$ (Aquila et al. (2014a)); -0.55 W/m$^2$ (Niemeier and Timmreck (2015)). In two cases, the forcing value was reported as the surface shortwave (SW) RF (Heckendorn et al.

(2009); Niemeier et al. (2011)): it has been converted to a net TOA RF by scaling the SW surface value with a factor (25-8)/20, where 25, 20 and 8 are the approximate factors to derive TOA SW, surface SW and TOA adjusted longwave (LW) RFs from the stratospheric AOD. From these RF values available in the literature, we may derive a mean value of -1.16 $\pm$ 0.33 W/m$^2$.

### 2.1.1 Changes in circulation and its feedback

While on the one hand these results show that SG leads to the desired effect of (at least partially) offsetting the positive RF of increasing well mixed greenhouse gases (WMGHG), on the other hand they show that SG effects, such as the lower stratospheric warming, must be carefully studied.

Enhanced lower stratospheric diabatic heating rates after major explosive volcanic eruptions and the consequent temperature increase were well documented both in observations (Labitzke and McCormick (1992); McLandress et al. (2015)) and through modeling experiments (Aquila et al. (2013); Pitari et al. (2016b)). The tropical lower stratospheric warming induces a significant increase of westerly winds from the thermal wind equation, with peaks at mid-latitudes in the mid-stratosphere. These dynamical changes tend to increase the amplitude of planetary waves in the stratosphere and to enhance the tropical upwelling in the rising branch of the Brewer Dobson circulation (BDC) (Pitari et al. (2014);Pitari et al. (2016a)).

The effects of the aerosol heating rates on the quasi-biennial oscillation (QBO) under geoengineering conditions have been analyzed in the aforementioned study by Aquila et al. (2014a) using the NASA Goddard chemistry-climate model (GEOSCCM), which includes an internally generated QBO. Four different experiments were designed, using 5 Tg-SO$_2$/yr for the first two and 2.5 Tg-SO$_2$/yr for the others, injected at different altitudes (16-25 km and 22-25 km; both at the equator and 0° longitude in a single lat/lon box). They found that SG perturbs the QBO by prolonging the westerly phase in the 20-50 hPa layer with an increasing stratospheric SO$_4$ mass burden (ranging from 1.5 Tg-S for the 16-25 km injection of 2.5 Tg-SO$_2$/yr to 4.7 Tg-S for the 22-25 km injection of 5 Tg-SO$_2$/yr).

Niemeier and Timmreck (2015) also mention a perturbation of the QBO in SG simulations performed with the ECHAM-HAM model. This was an ensemble of simulations with variable SO$_2$ injection (1-100 Tg-S/yr), altitude and latitude of injection (60 hPa and 30 hPa; Eq-2.8°N; 5°S-5°N; 30°S-30°N; all in a single longitude box centered at 122.3°E). Their simulations includes explicit aerosol microphysics, so that the effects of the perturbed QBO on the aerosol size distribution are taken into account. They found that an injection of about 8 Tg-S/yr would cause a slowing of the QBO oscillation with a constant QBO westerly phase in the lower stratosphere with overlaying easterlies, consistently with the findings by Aquila et al. (2014a). The overall conclusion of both these studies is that a stratospheric sulfur injection could dramatically alter the QBO periodicity, up to producing a permanent westerly phase in the lower stratosphere, thus reducing the meridional transport efficiency (Trepte and Hitchman (1992)).

The SO$_4$ stratospheric lifetime in the simulations included in Aquila et al. (2014a) was approximately 1.2 and 1.8 years for sulfur injection in the altitude layers 16-25 km and 22-25 km, respectively. However, it is interesting to note that the sulfate lifetime is systematically longer in the 5 Tg-SO$_2$/yr case with respect to the 2.5 Tg-SO$_2$/yr injection case ($\sim$1.9 years versus $\sim$1.7 years with injection in the 22-25 km layer and $\sim$1.25 years versus $\sim$1.2 years with injection in the 16-25 km layer).

The higher heating rates produced by the aerosol in the 5 Tg-SO$_2$/yr case are responsible for a stronger modification of the stratospheric circulation, resulting in the QBO changes and increased tropical upwelling, hence a better confinement of the particles in the tropical pipe (Trepte and Hitchman (1992); Pitari et al. (2016b)). This reduces the amount of aerosol that may be transported downwards across the extra-tropical tropopause in the lower branch of the BDC. A compact summary of all

these feedback mechanisms is presented in Fig. 2 (superimposed to the calculated sulfate mass density anomaly due to an injection of 5 Tg-SO$_2$/yr).

The prolonging of the aerosol lifetime found by Aquila et al. (2014a), however, could be canceled if the microphysical effects of the QBO-dependent sulfur confinement in the tropical pipe were taken into account. In the simulations by Niemeier and Timmreck (2015) using the ECHAM-HAM model, which includes a representation of aerosol microphysics, the enhanced

aerosol tropical confinement under condition of a locked QBO westerly phase in the lower stratosphere decreases the SG aerosol lifetime, this is because the tighter tropical confinement of the aerosol also leads to larger particles and therefore a more efficient gravitational settling (U. Niemeier, personal communications) (see Fig. 2b).

Many of the previous cited studies have focused on specific aspects of formation, transport and removal of stratospheric aerosols under geoengineering conditions. As noted above, significant feedback mechanisms exist among the magnitude and location

of SO$_2$ injection, aerosol microphysics, background stratospheric dynamics, aerosol induced surface cooling and stratospheric heating rates, as well as induced changes in the stratospheric circulation and strat/trop exchange. This means that a significant improvement on the knowledge of direct and indirect effects of SG may be obtained through model experiments designed in such a way that all these aspects are explicitly considered and interacting with each other. One important limitation of many of the above cited studies is the use of atmosphere-only models forced by prescribed sea surface temperatures (SST), so that an

explicit interaction of geoengineering aerosols with surface ocean is not considered. A missing explicit aerosol microphysics is another limitation for some of these studies: in this case, the increased gas-particle conversion cannot feedback on the aerosol size distribution shape and finally on the particle sedimentation rate and aerosol optical properties for the radiative transfer calculations.

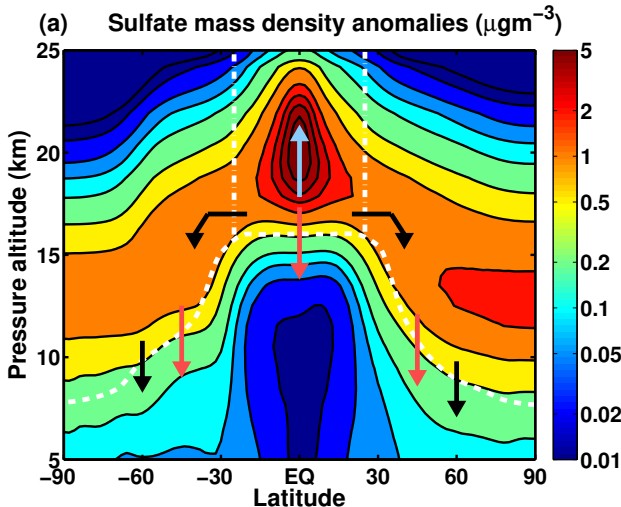

**(a)  Sulfate mass density anomalies ($\mu$gm$^{-3}$)**

**(b)  Summary of SO$_2$ injection feedback mechanisms**

| | Dynamical effect | → | With increasing SO$_2$ injection: | Sulfate lifetime & optical depth |
|---|---|---|---|---|
| ↓ | Gravitational settling | → | **Increases**<br>[Enhanced gas-particle conversion: larger particles] | **Decrease** |
| ⤺ ⤻ | Isentropic poleward transport & strat-trop exchange | → | **Decreases**<br>[Prolonged QBO W phase: higher tropical confinement] | **Increase** |
| ↓ | Tropical gravitational settling | → | **Increases**<br>[Higher sulfur confinement due to QBO effect: larger particles] | **Decrease** |
| ↑ | Tropical upwelling | → | **Increases**<br>[Enhanced aerosol heating rates] | **Increase** |

**Figure 2.** Panel (a): annually and zonally averaged sulfate mass density calculated anomalies ($\mu$g/m$^3$), due to a geoengineering injection of 5 Tg-SO$_2$/yr, with respect to a RCP4.5 background atmosphere. The aerosol mass density distribution is calculated in the Goddard Earth Observing System Chemistry Climate Model (GEOSCCM), with SG treated as described in Pitari et al. (2014). Arrows superimposed to the aerosol distribution indicate the main transport pathways of the aerosol particles, as explained in panel (b). The white dashed line shows the mean tropopause; the dash-dotted white lines highlight the stratospheric tropical region. The sensitivity of each dynamical effect to the SO$_2$ injection is highlighted in panel (b), along with the physical mechanisms driving the perturbation and the net effect on sulfate lifetime and optical depth.

## 2.2 Indirect radiative forcing

In the following subsections we shall summarize the indirect changes caused by the SG-induced stratospheric warming and surface cooling. This section answers the question if any of these indirect effects could significantly counteract or enhance the primary goal of SG of counteracting the positive RF from WMGHGs.

### 2.2.1 Ozone

Early studies of the potential impact of SG on stratospheric ozone are those of Tilmes et al. (2008), Tilmes et al. (2009) and Heckendorn et al. (2009). Tilmes et al. (2008) focus on polar ozone and estimate that SG could favor stratospheric ozone destruction and delay the recovery of the Antarctic ozone hole by 30-70 years. In addition, this ozone depletion produces a significant increase of erythemal surface UV, up to 5% in mid- and high latitudes and 10% over Antarctica (Tilmes et al. (2012)). The polar ozone depletion is favored by enhanced $NO_x$ removal via heterogeneous chemical reactions on the surface of stratospheric sulfate aerosols, as in the case of major volcanic eruptions taking place with high atmospheric levels of chlorine and bromine species (Tabazadeh et al. (2002)).

Tilmes et al. (2009) and Heckendorn et al. (2009) analyze the SG impact in chemical ozone loss rates and find that the chemical ozone changes are significantly impacted by the strong reduction of the $NO_x$ cycle, due to the efficient $NO_x$ to $HNO_3$ conversion on the surface of sulfate aerosols. The $NO_x$ depletion, in turn, favors an increase of $HO_x$, $Cl_x$ and $Br_x$ loss rates: the net effect on the ozone column will then be time-dependent and regulated by the amount of halogen species in the lower stratosphere. Heckendorn et al. (2009) have calculated a global ozone reduction of 4.5% (i.e., $\sim$13 DU), for an injection of 10 Tg-$SO_2$/yr and assuming halogen concentrations appropriate for the year 2000. Pitari et al. (2014) have run the GeoMIP G4 experiment from 2020 to 2070: despite the constant stratospheric aerosol loading, the magnitude of the geoengineering aerosol induced ozone depletion is found to decrease in time, due to the decreasing atmospheric concentration of chlorine and bromine species. Two of the models used in this study (ULAQ-CCM and MIROC-ESM-CHEM) even show a global ozone increase starting from about 2050, when the $NO_x$ driven chemical ozone increase is no longer over-balanced by the $HO_x$, $Cl_x$ and $Br_x$ driven ozone loss.

Model simulations in Pitari et al. (2014) showed that SG produces changes in stratospheric ozone due to a series of concurring factors, i.e., perturbation of photolysis rates because of the increased AOD, enhanced heterogeneous chemistry, and modifications of atmospheric dynamics. The models used in the G4 experiment show significant changes in the ozone profile, with a decrease in the tropical column between 100 and 30 hPa in the tropics, for the combined effects of enhanced upwelling and losses in the chemical cycles. Above that layer, ozone was found to increase because of the reduction of $NO_x$ via enhanced heterogeneous chemistry. Combined with similar changes in the extratropics, which are largely produced by modifications in the chemical processes, an average total change SG induced perturbation of -2.8$\pm$3.0 DU is calculated in the global mean ozone column, considering decadal averages from 2020 to 2070 for the 4 models that ran the G4 experiment (ULAQ-CCM, GEOSCCM, GISS-E2-R and MIROC-ESM-CHEM) and for the two models that ran the G3 experiment (ULAQ-CCM and GISS-E2-R). In terms of RF this produces a rather small negative result, of the order of -0.04 W/m$^2$ : RF=-0.045$\pm$0.035

W/m$^2$, with the same decadal averages used for the global mean ozone column change.

### 2.2.2 Stratospheric water vapor

SG is expected to increase stratospheric water vapor concentration by warming the TTL. In the stratosphere, the water vapor concentration is regulated by the TTL temperature (Dessler et al. (2013)), combined with methane oxidation. The higher the TTL temperatures, the more water vapor is able to enter the stratosphere. However, when considering the behavior of the TTL in a geoengineering scenario, we must consider two overlapping effects: an upper tropospheric cooling caused by the aerosol scattering, which cools the surface and stabilizes the troposphere (thus reducing convective heating), and a lower stratospheric warming caused by the infrared absorption by the aerosol particles. The amount of water vapor predicted in the stratosphere will thus depend on how the models represent these processes (Oman et al. (2008)).

Water vapor contributes to global warming, since it works as a GHG both in the troposphere and in the stratosphere (Forster F. and Shine (1999); Dessler et al. (2013)). Following the definition of radiative forcing, i.e., the net radiative flux change at the tropopause with fixed tropospheric temperatures and adjusted stratospheric temperatures, only stratospheric water vapor changes concur to the determination of the RF associated to any considered anthropogenic perturbation, SG in the present case. Pitari et al. (2014) gave an estimate of the RF of the SG-induced increase in stratospheric water vapor. At 100 hPa in the tropics, 3 out of 4 models produce a warming ranging from +0.16 K to +0.58 K that leads to an increase in water vapor mixing ratio from 0.08 to 0.35 ppmv. This in turn produces a net positive RF=0.055±0.025 W/m$^2$, considering decadal averages from 2020 to 2070 for the 3 of the 4 models that ran the G4 experiment (ULAQ-CCM, GEOSCCM, and MIROC-ESM-CHEM). The fourth model (GISS-E2-R), on the other hand, predicts a TTL cooling with a decreased amount of stratospheric H$_2$O and thus a negative RF. This is partly due to an underestimated lower stratospheric aerosol warming, originated by an insufficient tropical confinement of the aerosol cloud.

### 2.2.3 Upper tropospheric ice

Several studies have proposed mechanisms by which the SG would affect upper tropospheric cirrus clouds, reaching, however, contradictory conclusions. Cirisan et al. (2013) found that SG directly provides ice nuclei (IN) of a larger size with respect to those in the unperturbed atmosphere, resulting in a rather small increase in cirrus cloud coverage. Kuebbeler et al. (2012), on the other hand, found that SG would decrease cirrus cloud coverage because of changes in temperature, vertical velocity and water vapor updraft. The aerosol driven surface cooling, coupled with the lower stratospheric warming, stabilizes the atmosphere due to a decreased vertical temperature gradient, thus reducing the available turbulent kinetic energy and the vertical updraft (Karcher and Lohmann (2002); Lohmann and Karcher (2002)). This results in a decrease of the upper tropospheric ice crystals formation, which in turn produces a less efficient trapping of the planetary longwave radiation and a reduction of the net atmospheric greenhouse effect. Fig. 3 presents a compact summary of the dynamical perturbations induced by SG and relevant for the ice particle formation via homogeneous freezing. Lower vertical velocities force a decrease in ice crystals number

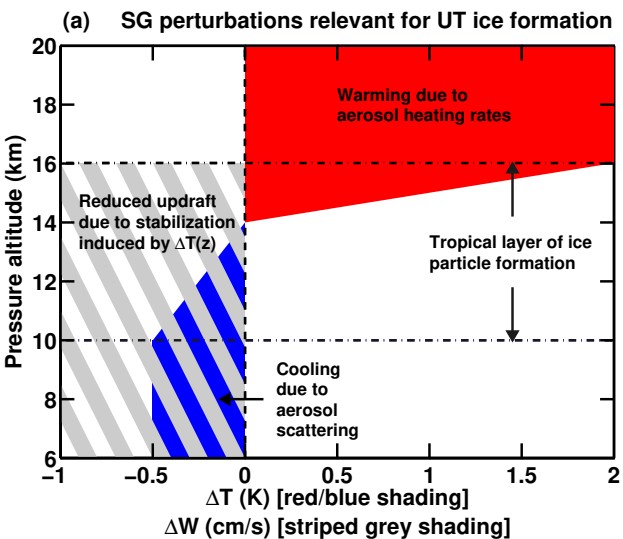

**(a)    SG perturbations relevant for UT ice formation**

**(b)    Summary of SO$_2$ injection feedback mechanisms**

| Thermal-Dynamical effect | With increasing SO$_2$ injection: | UT ice optical depth |
|---|---|---|
| Lower stratospheric & uppermost tropospheric warming | **Increases**<br>[Enhanced aerosol heating rates due to LW radiation absorption] | **Decreases**<br>[Faster depositional growth and lower nucleation rates] |
| Tropospheric cooling | **Increases**<br>[Enhanced aerosol SW radiation scattering] | **Increases**<br>[Slower depositional growth and higher nucleation rates] |
| Vertical velocity and water vapor updraft | **Decreases**<br>[Enhanced tropospheric stabilization due to induced T(z) changes] | **Decreases**<br>[Lower supersaturation: less ice crystals can nucleate] |
| Aerosol gravitational settling | **Increases**<br>[Enhanced gas-particle conversion: larger particles] | **Increases (?)**<br>[More UT sulfate aerosols, but inefficient IN for heterogeneous freezing] |

**Figure 3.** Panel (a): schematic profile changes of upper troposphere-lower stratosphere temperature (K) and UT vertical velocity (cm/s) in the tropics, due to a geoengineering injection of 5 Tg-SO$_2$/yr. The perturbation scheme is based on the findings of Kuebbeler et al. (2012), Pitari et al. (2016c) and Pitari et al. (2014). The dash-dotted black lines indicate the region of ice particle formation (up to the mean tropopause). The sensitivity of each thermal-dynamical effect to the SO$_2$ injection is highlighted in panel (b), along with the physical mechanisms driving the perturbation and the net effect on UT ice optical depth.

concentration due to the decreasing water vapor transport from below, with consequent lower supersaturation. The temperature dependence is inverse, because lower temperatures allow for more ice crystals, due to the slower depositional growth and the higher nucleation rate (Kuebbeler et al. (2012)).

### 2.2.3.1 Ice formation via homogeneous freezing

As clearly demonstrated in a number of papers focusing on the physical processes taking place in the upper troposphere (Karcher and Lohmann (2002); Hendricks et al. (2011)), the formation of ice particles may take place via heterogeneous and homogeneous freezing mechanisms. Airborne measurements by Strom et al. (1997) reported typical concentrations of newly formed ice crystals of the order of 0.3 cm$^{-3}$ in a young cirrus cloud at T=220 K in the upper troposphere of Northern Hemisphere mid-latitudes, in agreement with the model estimate of Karcher and Lohmann (2002) based on the assumption of ice particle formation via homogeneous freezing.

The homogeneous freezing mechanism normally dominates in the upper troposphere and involves water vapor freezing over liquid supercooled particles (as sulfate aerosols or sulfate coated aerosols), when the ice supersaturation ratio exceeds ∼1.5. In a SG perturbed atmosphere, more sulfate aerosols are available in the upper troposphere with respect to unperturbed background conditions thanks to extratropical downwelling and gravitational settling from the lower stratosphere. However, the background number density of sulfate aerosols in the upper troposphere is normally already much larger than the number of ice particles that can form (Karcher and Lohmann (2002)). This means that the SG driven increase of IN number density has basically no effect on the population of ice particles, but we may expect some impact on the ice particle size due to the larger size of IN made available by SG. This is the main conclusion of Cirisan et al. (2013), who note that the more large geoengineered particles exist (of typical sizes close to 0.5 $\mu$m), the less particles have to struggle against the Kelvin effect and more droplets may grow to larger sizes. This study analyzes in detail the direct SG impact on IN, as a complementary effect with respect to the dynamical indirect effect investigated by Kuebbeler et al. (2012). The main conclusion of Cirisan et al. (2013) is that the microphysical impact on cirrus clouds from geoengineered stratospheric sulfate aerosols is not an important side effect. They estimate a resulting mid-latitude average RF in the range of +0.02 W/m$^2$ to -0.04 W/m$^2$, depending on upwelling velocities and geoengineering scenario.This is consistent with the conclusions by Karcher and Lohmann (2002), who found that the effect of a perturbed aerosol size distribution on the ice particle population formed via homogeneous freezing is of secondary importance. It should be considered, however, that the estimates from Cirisan et al. (2013) are based on box model simulations and radiative transfer model calculations, and do not consider the dynamical impact and the feedback to microphysics.

### 2.2.3.2 Ice formation via heterogeneous freezing

The other possible pathway for ice crystal formation is through heterogeneous freezing, which requires solid nuclei as mineral dust or black carbon. In this case, when the ice supersaturation ratio exceeds approximately 1.1, heterogeneous freezing may start (Hendricks et al. (2011)); sulfate aerosols do not act as potential IN in this case. Kuebbeler et al. (2012) and, indirectly, Cirisan et al. (2013) have demonstrated that only the indirect dynamical perturbation induced by SG may be capable of significantly perturb the number density of upper tropospheric ice particles, with decreased vertical velocities due to the enhanced atmospheric stabilization. As noted in Kuebbeler et al. (2012), the idea proposed in some studies that volcanic eruptions may lead to enhanced ice crystal number concentrations was indeed confirmed by ISCCP lidar measurements (Sassen et al.

(2008)), whereas modeling studies found only a weak aerosol effect even in case of large perturbations (Karcher and Lohmann (2002); Lohmann and Feichter (2005)). However, it should be noted that in the case of explosive volcanic eruptions (contrary to SG) there are also solid ash particles injected in the lower stratosphere that will settle down below the tropopause (although with a rather short lifetime for the mass-dominant coarse mode), thus potentially contributing to some increase of the upper
tropospheric IN population actually available for heterogeneous freezing. Gettelman et al. (2010) have shown that mineral dust particles can play an important role in cirrus cloud formation, because their ice active fraction may be rather large (>10% for a supersaturation ratio close to the homogeneous freezing threshold). However, this is not the case for the proposed SG, where the homogeneous freezing mechanism actually dominates.

Recent studies by Storelvmo et al. (2013) and Storelvmo et al. (2014) have quantified the direct radiative effects produced by
seeding upper tropospheric cirrus ice clouds with large IN. Although this is not directly related to our specific discussion on SG side effects, it can be considered an indirect proof of the importance of correctly understanding the balance between the complex microphysical processes regulating the formation and growth of upper tropospheric ice particles.

### 2.2.3.3 RF estimates from cirrus ice thinning

We may conclude that the assumption of limiting our discussion to the indirect dynamical effect is a robust one and based on a sound physical basis. Kuebbeler et al. (2012) have calculated a LW TOA RF=-0.51 W/m$^2$ for cloud adjustment due to optically thinner cirrus, under a SG injection of 5 Tg-SO$_2$/yr. However, we should keep in mind that some degree of uncertainty remains for the processes regulating the potential direct perturbation of upper tropospheric ice crystals through changes in the
size distribution of sulfate aerosols acting as IN. In addition, as noted by the author themselves, one limitation of the study by Kuebbeler et al. (2012) is that sea surface temperatures were prescribed. The SG induced cooling of the surface would on one hand enhance the atmospheric stabilization and then further reduce the vertical updraft and cirrus ice optical depth (see Fig. 3), but on the other hand it would contribute to cool the whole troposphere, thus favoring additional ice crystals formation (see Fig. 3). Although Kuebbeler et al. (2012) suggest that in principle it would be important to redo the simulations with a mixed
layer ocean, on the other hand they conclude that the overall difference in the GCM response would be small in term of UT ice anomalies.

As shown in Pitari et al. (2016c) for the atmospheric stabilization resulting from tropospheric aerosols by non-explosive volcanoes, the combined effect of the aerosol induced tropospheric decrease in temperature and updraft velocities produces a net global reduction of ice optical thickness in the upper troposphere of $1.0 \times 10^{-3}$ at $\lambda$=0.55 $\mu$m, which then causes a radiative
forcing of -0.08 W/m$^2$. This corresponds to an aerosol optical depth increase of $5.3 \times 10^{-3}$ and an average surface cooling of 0.07 K. The same ULAQ-CCM module for ice crystals formation via homogeneous freezing has been applied to the SG case with stratospheric injection of 5 Tg-SO$_2$/yr, obtaining a globally averaged LW TOA RF=-0.45 W/m$^2$ due to optically thinner cirrus, consistent with the findings of Kuebbeler et al. (2012). A corresponding net TOA RF=-0.30 W/m$^2$ was calculated in all sky conditions , with the SW RF=0.15 W/m$^2$ (i.e., 34% of the absolute LW RF). If this same SW/LW RF fraction is applied, a

net TOA RF=-0.34 W/m$^2$ is obtained for Kuebbeler et al. (2012), for the cloud adjustment due to optically thinner cirrus.

### 2.2.4   Methane

Another indirect effect of SG is a lifetime modification for many long-lived species. Among these species CH$_4$ is particularly important, due to its sensitivity to OH abundance and its impact on tropospheric chemistry. A CH$_4$ lifetime increase takes place for three main reasons (Aquila et al. (2014b)), all connected with a decrease in OH concentration, which represents the main sink for methane: (a) the surface cooling directly lessens the amount of water vapor in the troposphere, which in turn diminishes the OH concentration. (b) A decrease in tropospheric UV occurs in the tropics because of the stratospheric aerosols. This reduces the production of O($^1$D), which in turns decreases the amount of OH produced by the reaction O($^1$D) + H$_2$O. (c) The increase of aerosol surface area density (SAD) enhances heterogeneous chemistry in the mid-upper troposphere, reducing the amount of NO$_x$ and O$_3$ production and thus of OH. The increased aerosol SAD produces a significant ozone depletion in the stratosphere, which results in an increase of UV radiation able to reach the surface. However, such effect is overbalanced by the direct scattering of solar radiation, so that the net amount of tropospheric UV is reduced (except over the polar latitudes) (Aquila et al. (2014b)). The high-latitude UV increase has little effect over the methane lifetime, which is mostly influenced from OH changes in the tropics.

In addition, it should be noted that the stratospheric aerosol heating rates produce a strengthening of the BDC, where more stratospheric air is transported from the stratosphere to the upper troposphere extra-tropics. Since the concentration of methane in the stratosphere is lower than in the troposphere, this strengthening of the BDC leads to a CH$_4$ decrease in the upper troposphere. All these effects together produce a longer lifetime of CH$_4$ that is estimated by the ULAQ-CCM to increase from 8 years for RCP4.5 to 9 years for SG with injection of 5 Tg-SO$_2$/yr. According to the model, such a lifetime increase is estimated to produce a positive TOA RF= +0.11 $\pm$ 0.04 W/m$^2$ (Aquila et al. (2014b)), as an average from year 2020 to 2090.

### 2.3   To what extent may SG balance WMGHG RF?

Here we discuss how the estimated net RF from direct and indirect effects of SG may be compared with the positive RF associated with increasing WMGHG. The current IPCC scenarios for the next century will produce by 2100 a RF relative to 2011 of 0.3 W/m$^2$ (RCP2.6), 2.2 W/m$^2$ (RCP4.5), 3.7 W/m$^2$ (RCP6.0) and 6.2 W/m$^2$ (RCP8.5) (IPCC (2013); Meinshausen et al. (2011)). In the subsequent discussion, we choose not to consider the most optimistic, but probably not realistic, scenario RCP2.6 with a sharp RF reduction already before 2100.

A total estimate of the net RF from SG must take into account the wide range of factors discussed in the previous subsections. Here we would like to highlight that the relationship between the SO$_2$ amount and the subsequent AOD is non-linear, as larger amounts of SO$_2$ will produce larger aerosol particles and the aerosol scattering efficiency decreases. Furthermore, the gravitational settling becomes faster with increasing particle size, therefore reducing the stratospheric aerosol lifetime.

As highlighted in sub-section 2.1, another factor that may change the aerosol lifetime is the prolonged QBO westerly phase caused by SG (Aquila et al. (2014a)). As showed by Pitari et al. (2016b) for explosive volcanic eruptions, a QBO with dom-

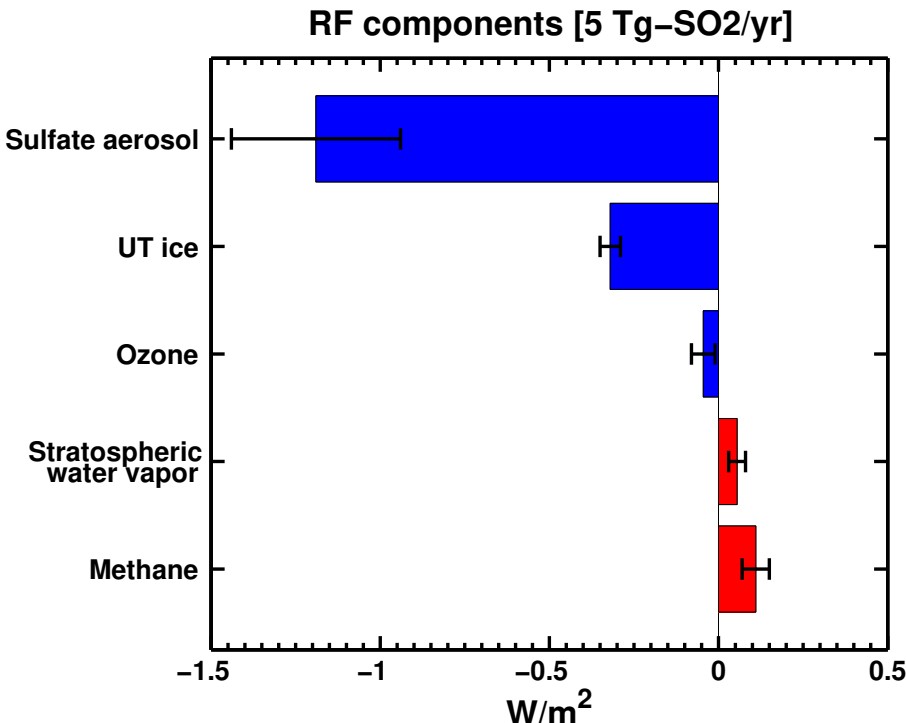

**Figure 4.** Summary of direct and indirect SG TOA RF per component (see subsections 2.1 - 2.2) (global mean values).

inant easterly shear leads to a longer lifetime for the volcanic aerosol, due to a greater isolation of the tropical pipe. This helps confining the aerosols in an area where downward transport is not present. In a similar way, the extension of the lower stratospheric QBO westerly phase simulated by Aquila et al. (2014a) leads to a longer aerosol lifetime. This result, however, could be partly canceled or even overcompensated if the microphysical effects of the QBO-dependent sulfur confinement in
5   the tropical pipe were taken into account. Niemeier and Timmreck (2015) found that a locked QBO westerly phase globally produces a net decrease of the SG aerosol lifetime, because the tropical isolation leads to larger particles and subsequently to a more efficient gravitational settling.

Figure 4 summarizes the RF breakdown per component, including direct and indirect effects of SG, as discussed in subsec-
10   tions 2.1 and 2.2 and based on published estimates. Aside from the direct effect of sulfate aerosol scattering, we see that the changes in UT ice particle formation and size may produce a significant negative RF, due to the thermal-dynamical induced thinning of cirrus clouds formed via homogeneous freezing. The indirect effects related to SG-induced changes in GHG concentrations ($CH_4$, $O_3$, stratospheric $H_2O$) are approximately one order of magnitude smaller, so that we may assume that they are globally negligible with respect to the direct effect of SG aerosols and their indirect impact on ice cloudiness.
15   Considering the results in Fig. 4, we find that the sum of all direct and indirect RFs of SG with an injection of 5 Tg-$SO_2$/yr

accounts for -1.4 $\pm$ 0.5 W/m$^2$, which means a compensation of the projected positive RF in 2100 relative to 2011 by 64%, 38% and 23% for the IPCC 'realistic' scenarios RCP4.5, RCP6.0 and RCP8.5, respectively. The November 2015 Paris Agreement aims to strengthen the global response to the threat of climate change by keeping a global temperature rise this century well below 2 $^\circ$C above pre-industrial levels and to pursue efforts to limit the temperature increase even further to 1.5 $^\circ$C. According to the IPCC (2013), the best estimate of the total anthropogenic RF relative to 1750 is 2.29 W/m$^2$ in 2011 and the increase in global mean surface temperature over the period 1880 to 2012 is 0.85 $^\circ$C. This means that the 2100 RF relative to 2011 projected in the three RCPs (2.2, 3.7 and 6.2 W/m$^2$, respectively) could not allow reaching the Paris Agreement target of a maximum temperature increase of $\sim$0.6 $^\circ$C up to $\sim$1.1 $^\circ$C in the period 2011 to 2100. In the hypothesis of SG implementation with injection of 5 Tg-SO$_2$/yr during the 21$^{st}$ century, the Paris Agreement target could likely be reached with the previously estimated SG-RF=-1.4$\pm$0.5 W/m$^2$. This could only happen in case of simultaneous WMGHG emissions regulated under scenario RCP4.5 or (barely) under scenario RCP6.0 (assuming a climate sensitivity of 0.5 K/Wm$^{-2}$).

## 3  Conclusions

Our assessment of the published literature on SG concludes that this proposed geoengineering technique has the potential to offset a significant part of the positive RF estimated during this century as a consequence of the increasing GHG concentrations. Both direct and indirect effects related to the stratospheric injection of 5 Tg-SO$_2$/yr need to be taken into account to produce robust conclusions. The rather large uncertainty in the direct sulfate forcing calculated from independent values available in the literature should not surprise, due to model differences in the treatment of aerosol microphysics, latitude and altitude of SO$_2$ injection, QBO effects, changes in large scale transport produced by the aerosol heating rates and surface cooling. The uncertainties still present could hopefully be reduced in future with multi-model results obtained from a wide array of global models in coordinated projects, such as GeoMIP, with strict specifications regarding the SO$_2$ injection and aerosol microphysics and transport.

Previous research works on SG have focused on specific aspects of formation, transport and removal of stratospheric aerosols under geoengineering conditions. However, significant feedback mechanisms exist among the magnitude and location of SO$_2$ injection, aerosol microphysics, background stratospheric dynamics, aerosol induced changes of SSTs, stratospheric heating rates and large scale circulation. For this reason, designing model simulations in which all these aspects are explicitly linked together is essential for producing more robust estimates of the direct and indirect effects of SG.

The net RF is considered here as a global average, providing no indication of how the regional climate would be effected by SG and how this would impact the hydrological cycle. Attention should also be used in studying the eventual side-effects of the termination of SG, so as to be sure that a powering down of the experiment would not have any negative side effect. Anyway, when comparing the SG techniques to others, it still appears to be one of the most feasible, taking into account its relatively high level of effectiveness and affordability (Robock et al. (2009); McClellan et al. (2012)). However, higher estimates on the SG costs have also been reported in the recent literature (Moriyama et al. (2016)), raising doubts on its affordability.

The above discussion highlights that still much is left to understand about the various effects on the climate of such a global

endeavour. In no way such studies have the goal of deciding whether such a task has to be carried out. That remains a prerogative of populations and decision-makers. What we can do is offer a deep insight on all possible consequences, if ever the need arises for any geoengineering method to be deployed.

5 *Acknowledgements.* The authors would like to thank U. Niemeier for helpful discussions.

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
