# Peer review of "Sulfate geoengineering: a review of the factors controlling the needed injection of sulfur dioxide"

_Atmospheric Chemistry and Physics, 2016_

## Referee Comment (RC1) · Anonymous Referee #1 · 18 Dec 2016

This is a review paper on sulfate geo-engineering and the factors controlling "the needed" injection of sulfur dioxide. The authors reviewed the direct radiative effect of sulfur injection that may lead to troposphere cooling and stratospheric warming, and the indirect radiative effect that caused by induced changes in ozone, CH4, stratospheric water vapor, and upper tropospheric cirrus clouds. They compared the effect of GHG warming and the resulted changes by the direct and indirect effects of sulfate geo-engineering in order to estimate the best amount of sulfate to be injected.

A critical review article that integrates and evaluates published literature is potentially very useful both for geo-engineering researchers and the broad atmospheric modeling community. Therefore, the effort the authors have made in this regard is greatly appreciated. However, I think the current manuscript needs to be substantially improved. The reasons are listed below:

1) A few sections (2.1, 2.2.1-2.2.3) in the current review only passively summarize the findings from previous studies, but they don't point out the weakness/gaps and suggest potential improvements and future directions. For example, many studies cited in the manuscript are based on the atmosphere-only model simulations forced by prescribed SST, so the interaction with the ocean is not considered. Another example is that the estimates from Cirisan et al. (2013) are based on box model simulations and radiative transfer model calculations, and it doesn't consider the dynamical impact and the feedback to microphysics. Some careful discussions are needed for such cases.

2) The authors did a good job in making connections between relevant studies, but in my opinion some of the discussions were presented with a bit too much detailed information (e.g. page 4 section 2.1), and the big picture was hided behind some mixed topics. For example, I would suggest the authors to divide section 2.1 (and possibly 2.2.x) into two parts: 1) direct effects of sulfur injection (changes in microphysical properties, aerosol lifecycle, and optical properties) and the associated heating and cooling; 2) changes in circulation and its feedback. Also, as a review article, I think it is necessary to draw some schematic plots showing the major findings (mechanisms) from the literature (e.g. one each for sections 2.1, 2.2.1-2.2.4), so that the readers can have a quick overview of those studies. This is particularly important when the authors want to deliver comprehensive messages and opposing points from different studies.

3) I think there are major flaws in Table 1 and the associated discussions (section 2.3). It seems to me that the authors are trying to project a net SG effect to compensate the RCP "forcing" (I think the authors should define their definition of forcing at the beginning) estimate. First, I am not clear how the authors derived the RCP RF numbers (not explicitly available in Moss et al. 2010), but it seems to me the "forcing" data presented in the paper are not calculated by CMIP models, but rather calculated using Integrated Assessment Models (IAM). Therefore, they might be very different from the real "forcing" estimated by the global climate models used in GeoMIP. Second, I think it's unacceptable to simply calculate the arithmetic mean the "forcing" numbers

obtained from studies on different (direct/indirect) SG effects and the RCP estimates. Even if these numbers are estimated from the same model, the non-linear effect between the GHG warming, sulfate scattering, and cirrus cloud formation would result a very different estimate. I suggest to eliminate this part.

4) Some additional literature need to be cited. For example, when discussing the impact on ozone, Tabazadeh et al. (2002) and Tilmes et al. (2008) should be cited and discussed.

References:

Tabazadeh, A., Drdla, K., Schoeberl, M.R., Hamill, P. and Toon, O.B., 2002. Arctic "ozone hole" in a cold volcanic stratosphere. Proceedings of the National Academy of Sciences, 99(5), pp.2609-2612.

Tilmes, S., Müller, R. and Salawitch, R., 2008. The sensitivity of polar ozone depletion to proposed geoengineering schemes. Science, 320(5880), pp.1201-1204.

Minor issues: I saw quite some formatting problems and typos (especially RCP numbers in table 1). Please correct them.

---

## Referee Comment (RC2) · Anonymous Referee #2 · 18 Dec 2016

Review

The paper summarizes geoengineering studies that discussed stratospheric SO2 injections into climate models. The paper focusses only on a few studies. There are not that many studies in recent years that actually injected a fixed amount of SO2 into the stratosphere. However, various studies used prescribed aerosol distributions. Those also contribute to the question of needed injections of sulfur dioxide. Therefore, I would recommend to extend this study to more papers, as listed below to justify the word "review" in the title. Also, I do not understand the last section of the paper and numbers in Table 1, and I think it needs more explanation. Specific comments are listed below:

Abstract: I disagree that the described technique would be planned for a timeframe of a few decades, while implementation of global measures of GHG emissions is achieved.

This technique would likely have to be applied during and after global measures are implemented, and for a much longer period of time if aiming for temperature stabilization, since temperatures will still continue to rise after mitigation efforts have started. See for example Sanderson et al., 2016 (doi: 10.1002/2016GL069563), Tilmes et al., 2016 (doi:10.1002/2016GL070122); depending on the mitigation efforts, solar geoengineering may be required for a very long period of time.

Line 10: It will be very difficult to fine-tune amounts of sulfur dioxide emissions based on models, due to the range of climate sensitivity and differences in the response of surface temperatures to volcanic aerosols. All the different studies can do, is outline important factors that control the amount of sulfur dioxide to be injected.

Page 2, Line 8. As commented above, it is misleading to assume that this technique would only be used between 2020 and 2070.

Page 2, Line 21. Why would you only focus on the G4 type studies, why not extend this? Besides, there are other earlier studies that used fixed amounts of SO2 injections, Rasch et al., 2006, and studies that prescribed sulfate aerosols based on fixed amounts of SO2 injections, including Rasch et al, 2008 (doi:10.1029/2007GL032179), Tilmes et al., 2009 (doi:10.1029/2008JD011420), Tilmes et al., 2012 (doi:10.5194/acp-12-10945-2012). Those and others may be included in the review.

Line 26: You can also add Niemeier et al., 2011 (doi:10.1002/asl.304), and Niemeier et al., 2013 (doi:10.1002/2013JD020445).

Page 3: Direct forcing of stratospheric sulfate: References in the first paragraph are very old and by now there are more recent papers describing that the cooling effect after Mt. Pinatubo was actually much smaller (at most 0.3 C), IPCC 2015, Canty et al., 2013 (doi:10.5194/acp-13-3997-2013). Also the radiative forcing seems to be largely overestimated in the study by Minnis et al., 1993.

Page 3, Line 28: The range in radiative response was likely due to the differences

in AOD of the models. However, even with the same AOD distribution, models may have very different radiative responses, see for example Neely et al., 2015 (doi:10.5194/gmdd-8-10711-2015), just comparing 2 CESM versions with different radiation schemes.

Page 3, Line 13: please change to "a series of factors"

Section 2.2.1 Ozone. This section only summarizes findings from one paper, this is not a review. Heckendorn et al., 2009 (doi:10.1088/1748- 9326/4/4/045108) and Tilmes et al., 2009 (doi:10.1029/2008JD011420), have discussed changes in ozone due to solar geoengineering.

Page 5, Line 13: Do the numbers -1.1 to -2.1DU include the model that did not consider heterogeneous chemistry? How do those numbers compare to earlier studies? Same for the RF, what models are included in this number?

Section 2.2.3. Do you mean "Upper tropospheric ice"?

Page 8, Line 12; Please note, tropospheric UV shows a net reduction in the tropics, correctly stated in the text. However, this is not the case of mid- and high latitudes. Methane lifetime is mostly influenced from OH changes in the tropics, therefore the methane lifetime is increased with geoengineering.

Line 23: typo: today's, also what do you mean by today's levels, what period?

Could you explain the numbers given in Section 2.3 and Table 1?

For example, to offset certain levels of RF, one would need to identify how much sulfur injection is required, which is model depended. For instance, Niemeier and Timmreck, 2015, calculated an efficiency of 0.30 – 0.35 W/m2 per TgS injection. Since 5TgSO2 are equal to 2.5 TgS, this results in about 0.3*2.5 = 0.75 W/m2 per 5 Tg SO2 injection. Can you do the same calculations for the other studies? It is not clear how you get to the value of -1.45 W/m2 +/- 0.65 in this study.

Also, for example the RF of RCP 4.5 between 2020 and 2070 is about 2.2-2.3 W/m2. Where does the number in Table 1 (0.8 W/m2) come from? If the RF needs to be set off by geoengineering in 2070, much more forcing is required than 0.8 W/m2.

For the cirrus forcing, why do you only state one number for cirrus impacts and not the lower number from Pitari et al., 2016b? Particle sizes from sulfate geoengineering are likely not large enough to have any significant effect, while dust particles have a larger effect. In Table 1, at least give a range for cirrus cloud effects.
* * *

---

## Referee Comment (RC3) · Anonymous Referee #3 · 29 Dec 2016

This paper presents a review of studies of sulfate geoengineering (SG). The paper selects results from a wide body of literature, of which some significant works have been left out. I agree with referee comment 1 (RC1) that more information should be given about the limitations of the studies presented, and the relative strength of conclusions possible. I have attached an annotated PDF with corrections and comments, which I summarize and expand on here.

Page 2, lines 7-8: This is a dangerously false statement. If SG were applied only during the transition period to clean energy source, its abrupt halt would trigger catastrophically rapid global warming, since the negative forcing of stratospheric sulfates would be removed within a few years, while the positive forcing of carbon dioxide would remain for thousands of years. Unless humans can remove much of the carbon dioxide from the atmosphere that we have added over 170 years thus far, SG would have to

be applied indefinitely on human timescales. As the paper alludes to, carbon air capture remains a very elusive and energy intensive process, and it is far from clear that it would be viable on a large scale by 2070.

Page 2, line 23: What is meant by "the GeoMIP experiment Robock et al. (2011)", in contrast to "the GeoMIP experiment G4" on line 21?

Page 3, line 6: It should be clarified that the 0.5°C drop in global average temperature was a monthly average, not an annual average.

Page 4, lines 7 and 12: clarify at what latitude(s) SO2 was injected, and how emissions were zonally distributed

Page 4, line 8: "proportionally" implies a linear relationship of aerosol mass injected to the period of the westerly phase. This does not see right if a permanent westerly is achieved with a finite injection rate.

Page 5, section 2.2.1: It is unclear that the attribution of reduction in O2 photolysis as the "main" cause of the reduction in column ozone is reasonable absent experiments in which O2 photolysis rates are unchanged by sulfate AOD. The catalytic loss rates are proportional to the amount of ozone present, so might be larger if ozone production were not reduced. The later discussion that column ozone increases with SG after 2060, when chlorine and bromine are reduced, makes this point less convincing.

I agree with RC2 that Table 1 is unclear and requires substantial further explanation.

I have included a few typographical corrections as well in the annotated PDF.

Finally, there are a number of additional studies that could be discussed in this review. RC1 and RC2 have identified a number of these. I would suggest at least including some discussion of these papers:

Tilmes, S., R. Müller, and R. Salawitch (2008), The sensitivity of polar ozone depletion to proposed geoengineering schemes, Science, 320(5880), 1201–1204,

doi:10.1126/science.1153966.

Tilmes, S. et al. (2013), The hydrological impact of geoengineering in the Geoengineering Model Intercomparison Project (GeoMIP), J Geophys Res-Atmos, 118(1), 11036–11058, doi:10.1002/jgrd.50868.

Tilmes, S., B. M. Sanderson, and B. C. O'Neill (2016), Climate impacts of geoengineering in a delayed mitigation scenario, Geophys Res Lett, 43(15), 8222–8229, doi:10.1002/2016GL070122.

Please also note the supplement to this comment:
http://www.atmos-chem-phys-discuss.net/acp-2016-985/acp-2016-985-RC3-supplement.pdf
* * *

---

## Author Comment (AC1) · 12 Jan 2017

Response to referee #1 attached as supplement.

Please also note the supplement to this comment:
http://www.atmos-chem-phys-discuss.net/acp-2016-985/acp-2016-985-AC1-supplement.pdf

[Figure]

[Figure]

**Figure 1.** Annual averaged vertical profiles of aerosol effective radius ($\mu$m) in the tropical stratosphere (25S-25N), with increasing geoengineering injection of $SO_2$ (see legend). The heavy dashed line indicates the mean tropical tropopause. Profiles are calculated in the University of L'Aquila Chemistry-Climate Model (ULAQ-CCM), which includes explicit gas-particle conversion and aerosol microphysics (Pitari et al. (2014)).

**Fig. 1.**

[Figure]

**(a)** Sulfate mass density anomalies ($\mu$gm$^{-3}$)

**(b)** Summary of SO$_2$ injection feedback mechanisms

| | Dynamical effect | With increasing SO$_2$ injection: | Sulfate lifetime & optical depth |
|---|---|---|---|
| ↓ | Gravitational settling | **Increases** [Enhanced gas-particle conversion: larger particles] | **Decrease** |
| ⟅⟆ | Isentropic poleward transport & strat-trop exchange | **Decreases** [Prolonged QBO W phase: higher tropical confinement] | **Increase** |
| ↓ | Tropical gravitational settling | **Increases** [Higher sulfur confinement due to QBO effect: larger particles] | **Decrease** |
| ↑ | Tropical upwelling | **Increases** [Enhanced aerosol heating rates] | **Increase** |

**Figure 2.** Panel (a): annually and zonally averaged sulfate mass density calculated anomalies ($\mu$g/m$^3$), due to a geoengineering injection of 5 Tg-SO$_2$/yr, with respect to a RCP4.5 background atmosphere. The aerosol mass density distribution is calculated in the Goddard Earth Observing System Chemistry Climate Model (GEOSCCM), with SG treated as described in Pitari et al. (2014). Arrows superimposed to the aerosol distribution indicate the main transport pathways of the aerosol particles, as explained in panel (b). The sensitivity of each dynamical effect to the SO$_2$ injection is highlighted in panel (b), along with the physical mechanisms driving the perturbation and the net effect on sulfate lifetime and optical depth.

8

**Fig. 2.**

**(a)   SG perturbations relevant for UT ice formation**

Warming due to aerosol heating rates

Reduced updraft due to stabilization induced by ΔT(z)

Tropical layer of ice particle formation

Cooling due to aerosol scattering

Pressure altitude (km)

ΔT (K) [red/blue shading]
ΔW (cm/s) [striped grey shading]

**(b)   Summary of SO₂ injection feedback mechanisms**

| Thermal-Dynamical effect | With increasing SO₂ injection: | UT ice optical depth |
|---|---|---|
| Lower stratospheric & uppermost tropospheric warming | Increases [Enhanced aerosol heating rates due to LW radiation absorption] | Decreases [Faster depositional growth and lower nucleation rates] |
| Tropospheric cooling | Increases [Enhanced aerosol SW radiation scattering] | Increases [Slower depositional growth and higher nucleation rates] |
| Vertical velocity and water vapor updraft | Decreases [Enhanced tropospheric stabilization due to induced T(z) changes] | Decreases [Lower supersaturation: less ice crystals can nucleate] |
| Aerosol gravitational settling | Increases [Enhanced gas-particle conversion: larger particles] | Increases (?) [More UT sulfate aerosols, but inefficient IN for heterogeneous freezing] |

**Figure 3.** Panel (a): schematic profile changes of upper troposphere-lower stratosphere temperature (K) and UT vertical velocity (cm/s) in the tropics, due to a geoengineering injection of 5 Tg-SO₂/yr. The perturbation scheme is based on the findings of Kuebbeler et al. (2012), Pitari et al. (2016c) and Pitari et al. (2014). The sensitivity of each thermal-dynamical effect to the SO₂ injection is highlighted in panel (b), along with the physical mechanisms driving the perturbation and the net effect on UT ice optical depth.

11

**Fig. 3.**

[Figure]

**Figure 4.** Summary of direct and indirect SG global TOA RF per component (see sections 2.1 - 2.2).

16

**Fig. 4.**

**Supplement:**

**Response to Reviewer 1 on "Sulfate geoengineering: a review of the factors controlling the needed injection of sulfur dioxide"**

***Comments are repeated in black italics.* Replies are indicated in blue. Figures 1, 2a, 2b, 3a, 3b and 4 have been attached.**

*This is a review paper on sulfate geo-engineering and the factors controlling "the needed" injection of sulfur dioxide. The authors reviewed the direct radiative effect of sulfur injection that may lead to troposphere cooling and stratospheric warming, and the indirect radiative effect that caused by induced changes in ozone, CH₄, stratospheric water vapor, and upper tropospheric cirrus clouds. They compared the effect of GHG warming and the resulted changes by the direct and indirect effects of sulfate geo-engineering in order to estimate the best amount of sulfate to be injected. A critical review article that integrates and evaluates published literature is potentially very useful both for geo-engineering researchers and the broad atmospheric modeling community. Therefore, the effort the authors have made in this regard is greatly appreciated. However, I think the current manuscript needs to be substantially improved.*

We thank the Reviewer for his encouraging general comment. As discussed below point-by-point, we have tried to incorporate all the Reviewer's suggestions for improving the manuscript.

*1) A few sections (2.1, 2.2.1-2.2.3) in the current review only passively summarize the findings from previous studies, but they don't point out the weakness/gaps and suggest potential improvements and future directions. For example, many studies cited in the manuscript are based on the atmosphere-only model simulations forced by prescribed SST, so the interaction with the ocean is not considered. Another example is that the estimates from Cirisan et al. (2013) are based on box model simulations and radiative transfer model calculations, and it doesn't consider the dynamical impact and the feedback to microphysics. Some careful discussions are needed for such cases.*

A caution statement has been included in section 2.1 specifying the limitations of many of the atmosphere-only model simulations. Suggestions for future directions are also included (in particular the full coupling of SG aerosol with climate, stratospheric heating rates, QBO and inclusion of explicit microphysics). A caution statement is also included in the discussion of Cirisan et al. (2013) estimates. Final recommendations are given in the conclusions.

*2) The authors did a good job in making connections between relevant studies, but in my opinion some of the discussions were presented with a bit too much detailed information (e.g. page 4 section 2.1), and the big picture was hidden behind some mixed topics. For example, I would suggest the authors to divide section 2.1 (and possibly 2.2.x) into two parts: 1) direct effects of sulfur injection (changes in microphysical properties, aerosol lifecycle, and optical properties) and the associated heating and cooling; 2) changes in circulation and its feedback. Also, as a review article, I think it is necessary to draw some schematic plots showing the major findings (mechanisms) from the literature (e.g. one each for sections 2.1, 2.2.1-2.2.4), so that the readers can have a quick overview of those studies. This is particularly important when the authors want to deliver comprehensive messages and opposing points from different studies.*

We have followed the reviewer suggestion by splitting up section 2.1, with an introductory part on the direct effects of sulfur injection and a subsection 2.1.1 on the changes in circulation and its feedback. We have also introduced schematic summary plots for three sections: Fig. 1 and Fig. 2a-b in section

2.1, Fig. 3a-b for section 2.2.3, Fig. 4 for section 2.3. Sub-section 2.2.3 has also been split in three parts (2.2.3.1, 2.2.3.2, 2.2.3.3) discussing separately the processes of ice formation via homogeneous and heterogeneous freezing and finally the estimates of RF due to cirrus ice thinning. The figures are attached to this response.

*3) I think there are major flaws in Table 1 and the associated discussions (section 2.3). It seems to me that the authors are trying to project a net SG effect to compensate the RCP "forcing" (I think the authors should define their definition of forcing at the beginning) estimate. First, I am not clear how the authors derived the RCP RF numbers (not explicitly available in Moss et al. 2010), but it seems to me the "forcing" data presented in the paper are not calculated by CMIP models, but rather calculated using Integrated Assessment Models (IAM). Therefore, they might be very different from the real "forcing" estimated by the global climate models used in GeoMIP. Second, I think it's unacceptable to simply calculate the arithmetic mean the "forcing" numbers obtained from studies on different (direct/indirect) SG effects and the RCP estimates. Even if these numbers are estimated from the same model, the non-linear effect between the GHG warming, sulfate scattering, and cirrus cloud formation would result a very different estimate. I suggest to eliminate this part.*

Section 2.3 has been reorganized and changed following the reviewer criticism. Table 1 and its discussion has been eliminated. We now present a summary of the RF values associated to SG that were previously discussed in sections 2.1 and 2.2, using values published in the literature.

*4) Some additional literature need to be cited. For example, when discussing the impact on ozone, Tabazadeh et al. (2002) and Tilmes et al. (2008) should be cited and discussed.*

*References:*

*Tabazadeh, A., Drdla, K., Schoeberl, M.R., Hamill, P. and Toon, O.B., 2002. Arctic "ozone hole" in a cold volcanic stratosphere. Proceedings of the National Academy of Sciences, 99(5), pp.2609-2612.*

*Tilmes, S., Müller, R. and Salawitch, R., 2008. The sensitivity of polar ozone depletion to proposed geoengineering schemes. Science, 320(5880), pp.1201-1204.*

The ozone impact section has been completed with additional citations of published articles, including the ones suggested by the reviewer.

*Minor issues: I saw quite some formatting problems and typos (especially RCP numbers in table 1). Please correct them.*

Table 1 has been eliminated (see comment above).

---

## Author Comment (AC2) · 12 Jan 2017

**Response to Reviewer 2 on "Sulfate geoengineering: a review of the factors controlling the needed injection of sulfur dioxide"**

***Comments are repeated in black italics.*** **Replies are indicated in blue. Figure 4 is attached to the response to reviewer 1.**

*The paper summarizes geoengineering studies that discussed stratospheric $SO_2$ injections into climate models. The paper focusses only on a few studies. There are not that many studies in recent years that actually injected a fixed amount of $SO_2$ into the stratosphere. However, various studies used prescribed aerosol distributions. Those also contribute to the question of needed injections of sulfur dioxide. Therefore, I would recommend to extend this study to more papers, as listed below to justify the word "review" in the title. Also, I do not understand the last section of the paper and numbers in Table 1, and I think it needs more explanation.*

We thank the Reviewer for his encouraging general comment. As discussed below point-by-point, we have tried to incorporate all the Reviewer's suggestions for improving the manuscript.

*Abstract: I disagree that the described technique would be planned for a timeframe of a few decades, while implementation of global measures of GHG emissions is achieved. This technique would likely have to be applied during and after global measures are implemented, and for a much longer period of time if aiming for temperature stabilization, since temperatures will still continue to rise after mitigation efforts have started. See for example Sanderson et al., 2016 (doi: 10.1002/2016GL069563), Tilmes et al., 2016 (doi:10.1002/2016GL070122); depending on the mitigation efforts, solar geoengineering may be required for a very long period of time.*

Both abstract and introduction have been modified according to this comment. The introduction now states: **"Such geoengineering methods would need to be applied during and after global intergovernmental 10 measures on GHG emissions are implemented, in order to achieve surface temperature stabilization (Sanderson et al. (2016); Tilmes et al. (2016))."**

*Line 10: It will be very difficult to fine-tune amounts of sulfur dioxide emissions based on models, due to the range of climate sensitivity and differences in the response of surface temperatures to volcanic aerosols. All the different studies can do, is outline important factors that control the amount of sulfur dioxide to be injected.*

Text modified according to this comment.

*Page 2, Line 8. As commented above, it is misleading to assume that this technique would only be used between 2020 and 2070.*

Text modified as suggested above.

*Page 2, Line 21. Why would you only focus on the G4 type studies, why not extend this? Besides, there are other earlier studies that used fixed amounts of $SO_2$ injections, Rasch et al., 2006, and studies that prescribed sulfate aerosols based on fixed amounts of $SO_2$ injections, including Rasch et al, 2008 (doi:10.1029/2007GL032179), Tilmes et al., 2009 (doi:10.1029/2008JD011420), Tilmes et al., 2012 (doi:10.5194/acp-12-10945-2012). Those and others may be included in the review.*

The reviewer suggestion has been followed in the revised version, including earlier studies with fixed amounts of $SO_2$ injections and also including a documented G3-type study in the ozone section.

*Line 26: You can also add Niemeier et al., 2011 (doi:10.1002/asl.304), and Niemeier et al., 2013 (doi:10.1002/2013JD020445).*

References added.

*Page 3: Direct forcing of stratospheric sulfate: References in the first paragraph are very old and by now there are more recent papers describing that the cooling effect after Mt. Pinatubo was actually much smaller (at most 0.3 C), IPCC 2015, Canty et al., 2013 (doi:10.5194/acp-13-3997-2013). Also the radiative forcing seems to be largely overestimated in the study by Minnis et al., 1993.*

References updated for the globally averaged temperature change after Pinatubo. The text now states: **"This was calculated as a monthly mean for September 1992, compared to pre-Pinatubo levels. However, more recent results with detrended analyses (Canty et al. (2013)) have shown that the Pinatubo volcanic impact on surface temperatures was probably overestimated by about a factor of 2, with a cooling estimate of 0.14 K and 0.32 K, globally and over land, respectively."** The estimate of Stowe et al. (1992) (~2.5 $Wm^{-2}$) is used for the net TOARF.

*Page 3, Line 28: The range in radiative response was likely due to the differences in AOD of the models. However, even with the same AOD distribution, models may have very different radiative responses, see for example Neely et al., 2015 (doi:10.5194/gmdd-8-10711-2015), just comparing 2 CESM versions with different radiation schemes.*

Text modified accordingly. We added the lines: **"The different results are mainly dependent on the (calculated, or imposed in one case) different aerosol optical depth (AOD) and size distribution among models. It should also be considered that, in general, even with the same AOD distribution, models may produce different radiative responses depending on the adopted radiation scheme (Neely et al. (2016))."**

*Page 3, Line 13: please change to "a series of factors".*

Changed.

*Section 2.2.1 Ozone. This section only summarizes findings from one paper, this is not a review. Heckendorn et al., 2009 (doi:10.1088/1748-9326/4/4/045108) and Tilmes et al., 2009 (doi:10.1029/2008JD011420), have discussed changes in ozone due to solar geoengineering.*

In the original manuscript we were focusing only on the topic of the indirect RF due to ozone changes, which was extensively reported only in Pitari et al. (2014). But we agree that in a review article the discussion should be extended to all relevant physical and chemical processes involved. A more complete coverage of the recent literature for the SG effects on stratospheric ozone is now made in the revised manuscript. We have added the following phrases: **"Early studies of the potential impact of SG on stratospheric ozone are those of Tilmes et al. (2008), Tilmes et al. (2009) and Heckendorn et al. (2009). Tilmes et al. (2008) focus on polar ozone and estimate that SG could favor stratospheric ozone destruction and delay the recovery of the Antarctic ozone hole by 30-70 years. In addition, this ozone depletion produces a significant increase of erythemal surface UV,**

up to 5% in mid- and high latitudes and 10% over Antarctica (Tilmes et al. (2012)). The polar ozone depletion is favored by enhanced NOx removal via heterogeneous chemical reactions on the surface of stratospheric sulfate aerosols, as in the case of major volcanic eruptions taking place with high atmospheric levels of chlorine and bromine species (Tabazadeh et al. (2002)). Tilmes et al. (2009) and Heckendorn et al. (2009) analyze the SG impact in chemical ozone loss rates and find that the chem- ical ozone changes are significantly impacted by the strong reduction of the NOx cycle, due to the efficient NOx to HNO3 conversion on the surface of sulfate aerosols. The NOx depletion, in turn, favors an increase of HOx, Clx and Brx loss rates: the net effect on column ozone column will then be time-dependent and regulated by the amount of halogen species in the lower stratosphere. Heckendorn et al. (2009) have calculated a global ozone reduction of 4.5% (i.e., ~13 DU), for an injection of 10 Tg-SO2/yr and assuming halogen concentrations appropriate for year 2000. Pitari et al. (2014) have run the GeoMIP G4 experiment from 2020 to 2070: despite the constant stratospheric aerosol loading, the magnitude of the geoengineering aerosol induced ozone depletion is found to decrease in time, due to the decreasing atmospheric concentration of chlorine and bromine species. Two of the models used in this study (ULAQ-CCM and MIROC-ESM-CHEM) even show a global ozone increase starting from about 2050, when the NOx driven chemical ozone increase is no longer over-balanced by the HOx, Clx and Brx driven ozone loss.”

*Page 5, Line 13: Do the numbers -1.1 to -2.1 DU include the model that did not consider heterogeneous chemistry? How do those numbers compare to earlier studies? Same for the RF, what models are included in this number?*

A more complete and precise discussion is now made in the revised manuscript, with appropriate citations to previous studies.

*Section 2.2.3. Do you mean “Upper tropospheric ice”?*

Thank you for catching this typo. Corrected.

*Page 8, Line 12; Please note, tropospheric UV shows a net reduction in the tropics, correctly stated in the text. However, this is not the case of mid- and high latitudes. Methane lifetime is mostly influenced from OH changes in the tropics, therefore the methane lifetime is increased with geoengineering.*

A sentence has been added to make it clear that the high-latitude UV increase has little effect on the methane lifetime.

*Line 23: typo: today’s, also what do you mean by today’s levels, what period? Could you explain the numbers given in Section 2.3 and Table 1? For example, to offset certain levels of RF, one would need to identify how much sulfur injection is required, which is model depended. For instance, Niemeier and Timmreck, 2015, calculated an efficiency of 0.30 – 0.35 $W/m^2$ per TgS injection. Since 5 Tg $SO_2$ are equal to 2.5 TgS, this results in about 0.3\*2.5 = 0.75 $W/m^2$ per 5 Tg $SO_2$ injection. Can you do the same calculations for the other studies? It is not clear how you get to the value of -1.45 $W/m^2$ +/- 0.65 in this study. Also, for example the RF of RCP 4.5 between 2020 and 2070 is about 2.2-2.3 $W/m^2$. Where does the number in Table 1 (0.8 W/m2) come from? If the RF needs to be set off by geoengineering in 2070, much more forcing is required than 0.8 $W/m^2$.*

Today's level is now specified: RF is estimated for year 2100 relative to year 2011. Table 1 has been eliminated. We agree that our attempt to quantify a net residual from the RCP net RFs over the "50 year period of SG application" minus the net RF from SG is not clear and not fully justified, on light of the previous criticisms. For this reason, we simply summarize the IPCC findings on the net RFs following different RCPs and we present our findings on the breakdown per component of the SG RF in a "stand-alone" figure, taking into account the estimates published in the recent literature and separately discussed in sections 2.1 and 2.2.

*For the cirrus forcing, why do you only state one number for cirrus impacts and not the lower number from Pitari et al., 2016b? Particle sizes from sulfate geoengineering are likely not large enough to have any significant effect, while dust particles have a larger effect. In Table 1, at least give a range for cirrus cloud effects.*

The RF summary plot (Fig. 4) now includes whiskers for all the components, including cirrus ice. We thank the reviewer for the specific suggestion. By the way, SG particles are inefficient IN, mainly because they are supercooled liquid particles, contrary to (solid) dust particles.

---

## Author Comment (AC3) · 12 Jan 2017

**Response to Reviewer 3 on "Sulfate geoengineering: a review of the factors controlling the needed injection of sulfur dioxide"**

***Comments are repeated in black italics.*** **Replies are indicated in blue.**

*This paper presents a review of studies of sulfate geoengineering (SG). The paper selects results from a wide body of literature, of which some significant works have been left out. I agree with referee comment 1 (RC1) that more information should be given about the limitations of the studies presented, and the relative strength of conclusions possible. I have attached an annotated PDF with corrections and comments, which I summarize and expand on here.*

We thank the Reviewer for his constructive comments. As discussed below point-by-point, we have tried to incorporate all the Reviewer's suggestions for improving the manuscript.

*Page 2, lines 7-8: This is a dangerously false statement. If SG were applied only during the transition period to clean energy source, its abrupt halt would trigger catastrophically rapid global warming, since the negative forcing of stratospheric sulfates would be removed within a few years, while the positive forcing of carbon dioxide would remain for thousands of years. Unless humans can remove much of the carbon dioxide from the atmosphere that we have added over 170 years thus far, SG would have to be applied indefinitely on human timescales. As the paper alludes to, carbon air capture remains a very elusive and energy intensive process, and it is far from clear that it would be viable on a large scale by 2070.*

Following the same recommendation of the second reviewer, we have cut this statement from both introduction and abstract.

*Page 2, line 23: What is meant by "the GeoMIP experiment Robock et al. (2011)", in contrast to "the GeoMIP experiment G4" on line 21?*

That was a typo: "G3" is missing. Corrected, with additional references.

*Page 3, line 6: It should be clarified that the 0.5 C drop in global average temperature was a monthly average, not an annual average.*

Following the same criticism by the second reviewer, we have corrected this statement, with additional references.

*Page 4, lines 7 and 12: clarify at what latitude(s) SO$_2$ was injected, and how emissions were zonally distributed.*

Clarified, for both Aquila et al. (2014) and Niemeier and Timmreck (2015).

*Page 4, line 8: "proportionally" implies a linear relationship of aerosol mass injected to the period of the westerly phase. This does not see right if a permanent westerly is achieved with a finite injection rate.*

We agree that "proportional" is not the right word to describe this effect. Corrected as follow: **"They found that an injection of about 8Tg-S/yr would cause a slowing of the QBO oscillation with a**

**constant QBO westerly phase in the lower stratosphere with overlaying easterlies, consistently with the findings by Aquila et al. (2014a).”**

*Page 5, section 2.2.1: It is unclear that the attribution of reduction in $O_2$ photolysis as the "main" cause of the reduction in column ozone is reasonable absent experiments in which $O_2$ photolysis rates are unchanged by sulfate AOD. The catalytic loss rates are proportional to the amount of ozone present, so might be larger if ozone production were not reduced. The later discussion that column ozone increases with SG after 2060, when chlorine and bromine are reduced, makes this point less convincing.*

We admit there was some confusing statements in the original manuscript. We have simplified our sentence as follows: **"The models used in the G4 experiment showed significant changes in the ozone profile, with a decrease in the tropical column between 100 and 30 hPa in the tropics, for the combined effects of enhanced upwelling and losses in the chemical cycles."**

*I agree with RC2 that Table 1 is unclear and requires substantial further explanation.*

Table 1 has been eliminated. We agree that our attempt to quantify a net residual from the RCP net RFs over the "50 year period of SG application" minus the net RF from SG is not clear and not fully justified, on light of the previous criticisms. For this reason we simply summarize the IPCC findings on the net RFs following different RCPs and we present our findings on the breakdown per component of the SG RF in a "stand-alone" figure, taking into account the estimates published in the recent literature and separately discussed in sections 2.1 and 2.2.

*I have included a few typographical corrections as well in the annotated PDF.*

The sticky notes on the original pdf document have been properly considered in the revised manuscript.

*Finally, there are a number of additional studies that could be discussed in this review. RC1 and RC2 have identified a number of these. I would suggest at least including some discussion of these papers:*

*Tilmes, S., R. Müller, and R. Salawitch (2008), The sensitivity of polar ozone depletion to proposed geoengineering schemes, Science, 320(5880), 1201–1204, doi:10.1126/science.1153966.*

*Tilmes, S. et al. (2013), The hydrological impact of geoengineering in the Geoengineering Model Intercomparison Project (GeoMIP), J. Geophys. Res-Atmos, 118(1), 11036–11058, doi:10.1002/jgrd.50868.*

*Tilmes, S., B. M. Sanderson, and B. C. O'Neill (2016), Climate impacts of geoengineering in a delayed mitigation scenario, Geophys. Res. Lett., 43(15), 8222–8229, doi:10.1002/2016GL070122.*

These (and other references to relevant SG studies) are included in the revised manuscript.

*Please also note the supplement to this comment: http://www.atmos-chem-phys-discuss.net/acp-2016-985/acp-2016-985-RC3-supplement.pdf*

The sticky notes on the original pdf document have been properly considered in the revised manuscript.

---

## Author Response (AR2)

**Response to the Reviewer.**

*Replies are in italics.*

Second round comment on "Sulfate geoengineering: a review of the factors controlling the needed injection of sulfur dioxide" by Daniele Visioni et al.

The new version of the paper by Daniele Visioni and co-authors has improved compared to the previous one, and the suggestions of changes based the reviews were to the most part implemented.

I only have one main concern regarding Section 2.3, which was added to this paper in the new version, and which needs to be revised before publication. The authors estimate the RF forcing components, based in a 5 TgSO2 per year injection rate. However, they go a step further and extrapolate the amount of SO2 injection that is required to offset the forcing by the end of the 21st century, with the goal to reach the 2 degree temperature target. However, their values do not agree with what is published for example in Niemeier and Timmreck, 2015. As stated in Niemeier and Timmreck, 2015, about 0.1-0.2 W/m2 can be reached per 1Tg sulfur injection. These values suggest that a reduction in RF of 2.2W/m2 would require injections of at least 12 or more TgS per year (or 24TgSO2 /yr). This is much more that what is assumed in the calculations in this paper and therefore should be addressed. Regarding the last sentence in section 2.3, why do the authors support the statement that 1.4 W/m2 is the amount that could be reached using stratospheric SO2 injections, and not more or less? Doesn't it depend on the amount of injection? Consequently, the first sentence of the conclusions needs to be revised.

*We thank the Reviewer for pointing out that the last sentence in Section 2.3 needs to clearly state that the SG related TOARF=-1.4 W/m² obtained from independent scientific studies is associated to a SG implementation **with 5 Tg-SO₂/yr during the 21ˢᵗ century**. The last sentence in Section 2.3 has been modified as follows:" In the hypothesis of SG implementation with injection of 5 Tg-SO₂/yr during the 21ˢᵗ century, the Paris Agreement target could likely be reached with the previously estimated SG RF=-1.4±0.4 W/m², in case of simultaneous WMGHG emissions regulated under scenario RCP4.5 or (barely) under scenario RCP6.0 (assuming a climate sensitivity of 0.5 K/Wm⁻²)." This is the injection amount to which our review analysis has been based. Once the injection amount has been clearly specified, the first sentence in the conclusions is correct.*

*The reviewer comment was also helpful to us for correcting an error made at the end of section 2.1. In fact, using Eq. (1) in the paper of Niemeier and Timmreck (2015), the net TOARF corresponding to 2.5 Tg-S/yr (i.e., 5 Tg-SO₂/yr) is -0.55 W/m², not -0.87 as we originally stated in the manuscript. This makes the overall averaged RF=-1.16±0.33 W/m², instead of -1.19±0.27.*

*The first criticism, related to a comparison of a 2.2 W/m² TOARF unbalance with findings from Niemeier and Timmreck (2015) is somewhat wrong, in our opinion. Our analysis on the potential*

*impact of SG with 5 Tg-$SO_2$/yr during the 21$^{st}$ century comes up with an average TOARF correction of - 1.4±0.4 W/m$^2$ to the 2.2 W/m$^2$ unbalance in 2100 with respect to 2011, assuming the RCP4.5 scenario. This means that a net increase of 0.8±0.4 W/m$^2$ would be obtained in the 2100 TOARF with respect to year 2011, corresponding to a globally averaged surface temperature warming of 0.4±0.2°C (assuming a climate sensitivity of 0.5 K/Wm$^{-2}$). Summing up this temperature change with the IPCC estimate of a 0.85 °C warming in 2012 with respect to 1880 conditions, we conclude that under the RCP4.5 scenario coupled to a 5 Tg-$SO_2$/yr SG implementation, the 1.5 °C target of the Paris agreement could indeed be reached. In the RCP6.0 scenario assumption the TOARF unbalance in 2100 with respect to 2011 would raise to 3.7 W/m$^2$, leaving a net of 2.3±0.4 W/m$^2$ in case of a 5 Tg-$SO_2$/yr SG implementation, i.e., a globally averaged surface temperature warming of 1.15±0.2°C. Summing up this temperature change with the IPCC estimate of a 0.85 °C warming in 2012 with respect to 1880 conditions, we conclude that under the RCP6.0 scenario coupled to a 5 Tg-$SO_2$/yr SG implementation, the 2 °C target of the Paris agreement could be reached, although the most desired 1.5 °C target would be missed.*

[revised manuscript text omitted]